# Navigating the Landscape of Liquid Biopsy in Colorectal Cancer: Current Insights and Future Directions

**DOI:** 10.3390/ijms26157619

**Published:** 2025-08-06

**Authors:** Pina Ziranu, Andrea Pretta, Giorgio Saba, Dario Spanu, Clelia Donisi, Paolo Albino Ferrari, Flaviana Cau, Alessandra Pia D’Agata, Monica Piras, Stefano Mariani, Marco Puzzoni, Valeria Pusceddu, Ferdinando Coghe, Gavino Faa, Mario Scartozzi

**Affiliations:** 1Medical Oncology Unit, University Hospital and University of Cagliari, 09042 Cagliari, Italy; an.pretta@gmail.com (A.P.); sabagiorgio@live.it (G.S.); dario.spanu@gmail.com (D.S.); cleliadonisi@gmail.com (C.D.); alessandrapiadagata@gmail.com (A.P.D.); mariani.step@gmail.com (S.M.); marcopuzzoni@gmail.com (M.P.); valeria.pusce@gmail.com (V.P.); marioscartozzi@gmail.com (M.S.); 2Division of Thoracic Surgery, Azienda di Rilievo Nazionale ed Alta Specializzazione “G. Brotzu”, 09121 Cagliari, Italy; paoloalb.ferrari@gmail.com; 3Department of Medical Sciences and Public Health, University of Cagliari, 09042 Cagliari, Italy; flacau@tiscali.it (F.C.); monica.piras@unica.it (M.P.); gavinofaa@gmail.com (G.F.); 4Clinical-Microbiological Laboratory, University Hospital of Cagliari, 09042 Cagliari, Italy; coghe.f@tiscali.it; 5Department of Biology, College of Science and Technology, Temple University, Philadelphia, PA 19122, USA

**Keywords:** liquid biopsy, colorectal cancer, minimal residual disease, resistance mechanisms, adjuvant therapy guidance

## Abstract

Liquid biopsy has emerged as a valuable tool for the detection and monitoring of colorectal cancer (CRC), providing minimally invasive insights into tumor biology through circulating biomarkers such as circulating tumor DNA (ctDNA), circulating tumor cells (CTCs), microRNAs (miRNAs), long non-coding RNAs (lncRNAs), and circular RNAs (circRNAs). Additional biomarkers, including tumor-educated platelets (TEPs) and exosomal RNAs, offer further potential for early detection and prognostic role, although ongoing clinical validation is still needed. This review summarizes the current evidence on the diagnostic, prognostic, and predictive capabilities of liquid biopsy in both metastatic and non-metastatic CRC. In the non-metastatic setting, liquid biopsy is gaining traction in early detection through screening and in identifying minimal residual disease (MRD), potentially guiding adjuvant treatment and reducing overtreatment. In contrast, liquid biopsy is more established in metastatic CRC for monitoring treatment responses, clonal evolution, and mechanisms of resistance. The integration of ctDNA-guided treatment algorithms into clinical practice could optimize therapeutic strategies and minimize unnecessary interventions. Despite promising advances, challenges remain in assay standardization, early-stage sensitivity, and the integration of multi-omic data for comprehensive tumor profiling. Future efforts should focus on enhancing the sensitivity of liquid biopsy platforms, validating emerging biomarkers, and expanding multi-omic approaches to support more targeted and personalized treatment strategies across CRC stages.

## 1. Introduction

Colorectal cancer (CRC) is among the most prevalent gastrointestinal malignancies and represents the second leading cause of cancer-related mortality worldwide [1,2,3]. Lifestyle factors such as poor diet and physical inactivity significantly contribute to its global burden [4]. CRC typically arises through the stepwise progression of colorectal adenomas to invasive carcinoma, often resulting in late-stage diagnosis and limited treatment options [5]. Prognosis is stage-dependent, with five-year survival rates decreasing from over 90% in stage I to 14% in stage IV [6,7,8], emphasizing the need for effective early detection strategies.

Conventional diagnostic methods, such as colonoscopy, imaging, biopsy with histological examination, and tumor markers, face limitations including invasiveness, limited sensitivity in early disease, and challenges related to tumor heterogeneity. Non-invasive tools such as the fecal immunochemical test (FIT) offer greater accessibility but lack adequate sensitivity for early-stage CRC [9]. Novel stool-based biomarkers, including DNA [10], mRNA [11], proteins [12], and microbiota profiles [13], are under development, yet their clinical utility remains constrained by high costs, variable adherence, and limited sensitivity for precancerous lesions.

Liquid biopsy has emerged as a promising non-invasive modality for tumor detection and monitoring. It enables the analysis of circulating tumor cells (CTCs), circulating tumor DNA (ctDNA), tumor-educated platelets (TEPs), exosomes, and tumor-associated proteins from blood or other body fluids [14,15]. This approach supports repeated sampling, real-time molecular profiling, and longitudinal assessment of tumor dynamics, aligning with the principles of precision oncology. In CRC, liquid biopsy holds significant potential for early detection, post-surgical surveillance, and individualized treatment planning, especially in patients with unresectable disease. It also offers valuable insights into therapeutic response, resistance mechanisms, and disease progression (Figure 1).

This review explores the application of liquid biopsy in CRC, focusing on its biological relevance, detection technologies, areas of applications, current limitations, and future clinical integration.

### 1.1. Historical Development of Liquid Biopsy: From Discovery to Clinical Translation

The evolution of liquid biopsy can be categorized into four key phases: the exploratory stage (pre-1990s), foundational research (1990s), technological and industrial expansion (2000–2010), and clinical translation and integration (2010–present) [16].

In the exploratory phase, pivotal discoveries laid the groundwork for modern liquid biopsy. In 1869, Ashworth identified malignant cell-like elements in the blood, now recognized as CTC [17]. In 1948, Mandel and Métais detected cell-free DNA (cfDNA) in plasma [18], while Wolf provided the first electron micrographs of extracellular vesicles (EVs) in 1967 [19]. In the early 1980s, Johnstone et al. described exosome biogenesis [20], and in 1977 Leon et al. found elevated cfDNA levels in cancer patients, linking them to tumor burden [21].

Refined methodologies, with polymerase chain reaction (PCR), allowed the detection of KRAS mutations in cfDNA from pancreatic cancer patients in 1994 [22]. In 1996, Raposo et al. demonstrated antigen presentation by EVs from immune cells [23], and in 1998 CTC isolation was correlated with tumor stage and prognosis [24].

The early 2000s brought significant technological advances. In 2005, CTC enumeration predicted survival in metastatic breast cancer [25]. In 2008, Diehl et al. used BEAMing (beads, emulsion, amplification, and magnetics) to track ctDNA mutations in CRC, correlating them with tumor burden and carcinoembryonic antigen (CEA) serum levels [26].

Since 2010, clinical applications have expanded. In 2014, the European Medicines Agency (EMA) approved plasma ctDNA testing for epidermal growth factor receptor (EGFR) mutations in non-small cell lung cancer (NSCLC) [27], followed by the U.S. Food and Drug Administration (FDA) approval of the Epi proColon^®^ (Berlin, Germany) test for CRC screening in 2016 [28]. The FDA also authorized ctDNA-based PIK3CA testing for hormone receptor-positive/human epidermal growth factor receptor 2-negative (HR^+^/HER2^−^) breast cancer [29] and BRCA1/2 mutation detection for poly ADP-ribose polymerase (PARP) inhibitor use in metastatic castration-resistant prostate cancer (mCRPC) [30].

Despite regulatory limitations, clinical use of liquid biopsy continues to grow, driven by robust translational research and ongoing clinical trials.

### 1.2. Molecular Biomarkers in Liquid Biopsy: Features, Advantages, Limitations, and Detection Techniques

Liquid biopsy provides a minimally invasive method for obtaining molecular data from blood, urine, and saliva in real time, encompassing CTCs, ctDNA, exosomes, TEPs, and non-coding RNAs (ncRNAs), including microRNAs (miRNAs), long non-coding RNAs (lncRNAs), and circular RNAs (circRNAs) [31,32,33,34,35,36,37,38,39,40,41,42,43,44,45,46,47,48,49,50,51,52,53,54,55,56,57,58,59,60,61,62,63,64,65,66,67,68,69,70,71,72,73,74,75,76,77,78,79,80,81,82,83,84,85,86,87,88,89,90,91,92,93,94,95,96,97]. A comparative summary of the main circulating biomarkers, including detection methods, sensitivity, specificity, and clinical applications, is provided in Table 1.

CTCs, first identified by Ashworth in 1869 [31], are tumor cells that detach from primary or metastatic lesions and enter the bloodstream, correlating with metastasis and prognosis in cancers such as breast and C [32,33,34,35]. Due to their low concentration (less than 10 cells per 10 mL of blood) and heterogeneity, detection methods involve enrichment and immunofluorescent labeling [36,37]. The CellSearch^®^ system, the only U.S. FDA-approved platform for CTC detection, uses epithelial cell adhesion molecule (EpCAM)-coated beads, but it may miss CTCs undergoing epithelial–mesenchymal transition (EMT), affecting sensitivity [38,39,40,41,42,43,44].

ctDNA, a small fraction of tumor-derived DNA, reflects the tumor’s mutational landscape and constitutes 0.1–1% of cfDNA [45,46,47]. Detection of ctDNA involves techniques such as digital droplet PCR (ddPCR), next-generation sequencing (NGS), and quantitative PCR (qPCR) [48,49,50,51,52]. However, ctDNA detection is challenging in patients with low tumor burden, and its fragmented nature may increase false negatives [53,54,55,56].

Exosomes, extracellular vesicles (30–150 nm), carry proteins, nucleic acids and lipids, reflecting the tumor microenvironment (TME) and protecting cargo from enzymatic degradation [57,58,59,60,61]. Isolation methods include ultracentrifugation, size-exclusion chromatography, and microfluidics, though protocol variability affects consistency [62,63,64,65,66,67,68].

TEPs absorb RNAs, proteins and vesicles from tumor cells, with RNA profiles potentially indicating tumor type, location, and mutation status [69,70]. RNA sequencing (RNA-seq) is the primary detection method but, despite promising accuracy in distinguishing cancer patients from controls, TEPs remain investigational and lack FDA-approved assays [71,72,73,74].

ncRNAs, including miRNAs, lncRNAs, and circRNAs, are emerging biomarkers [75,76,77,78,79,80,81,82,83,84,85,86,87,88,89,90,91,92,93,94,95,96]. miRNAs (18–25 nucleotides) regulate gene expression and are detected via qPCR, but hemolysis may impact results [75,76]. lncRNAs (over 200 nucleotides) regulate gene expression and chromatin structure, detectable via RNA fluorescence in situ hybridization (RNA-FISH), reverse transcription qPCR (RT-qPCR), and RNA-seq, though their variability complicates interpretation [77,78,79,80,81,82,83,84,85,86,87,88,89]. circRNAs (200–600 nucleotides), characterized by a covalently closed-loop structure, act as miRNA sponges and transcriptional regulators and are detectable by divergent primer RT-PCR and RNA-seq, though standardization is still needed [90,91,92,93,94,95,96,97].

Despite advancements, liquid biopsy faces challenges in standardization, sensitivity, and cost. Further research is necessary to optimize detection methods, validate biomarker specificity, and establish clinical guidelines to enable broader clinical adoption.

### 1.3. Clinical Applications of Liquid Biopsy in Colorectal Cancer

In recent years, liquid biopsy has rapidly evolved from a promising research tool to a clinically relevant strategy in the management of CRC [98]. Its minimally invasive nature, combined with the ability to provide dynamic molecular insights, offers significant advantages over traditional tissue-based approaches. In the next chapters, we explore the key clinical applications of liquid biopsy across three major settings of CRC management: screening, where early detection can significantly impact patient outcomes; localized disease, where liquid biopsy can assist in risk stratification, detection of MRD, and guidance of adjuvant therapy; and metastatic disease, where it plays a crucial role in treatment selection, monitoring response, and identifying mechanisms of resistance. Together, these applications underscore the transformative potential of liquid biopsy in achieving more personalized and adaptive care in CRC (Figure 1).

## 2. Screening

The primary goal of CRC screening is early diagnosis to improve patient outcomes. Commonly used methods include FIT, which has a sensitivity of 74% and specificity of 96%; the high-sensitivity guaiac-based fecal occult blood test (HSgFOBT), with 70% sensitivity and 93% specificity; and colonoscopy which offers 95% sensitivity and 86% specificity [99]. However, these methods have limitations, particularly the lower sensitivity of FIT and HSgFOBT and the low patient compliance with invasive procedures like colonoscopy [99,100,101]. Developing less invasive, more accessible diagnostic tests remains a key clinical need [102]. Liquid biopsy, detecting circulating biomarkers in blood, offers a potential solution to overcome these limitations [101]. The role of CTCs in the early diagnosis of CRC remains limited, due to the low sensitivity observed in several studies [103,104]. Therefore, this chapter will primarily focus on the role of circulating nucleic acids and other biomarkers (Table 2).

### 2.1. Circulating Cell-Free DNA and Circulating Tumor DNA

Circulating cfDNA, released into the bloodstream through apoptosis and necrosis, has emerged as a key tool for molecular assessment of CRC and other cancers [135,136,137]. A small fraction of cfDNA, known as ctDNA, carries tumor-specific genetic alterations, including somatic mutations, structural variants, copy-number changes, and microsatellite instability, enabling non-invasive tumor monitoring [138]. The potential of ctDNA testing lies in detecting malignancy-associated genetic signatures, addressing limitations of conventional CRC screening [139]. However, in early-stage CRC, low ctDNA levels increase the risk of false negatives [140]. Mutational heterogeneity further complicates ctDNA analysis, as certain mutations may not be universally present [141]. To enhance sensitivity, emerging approaches focus on DNA methylation markers, fragmentomics, and proteomics to detect even minimal tumor DNA more effectively [140].

The EpiProColon^®^ assay, FDA-approved in 2016, targets SEPT9 gene methylation for CRC screening, achieving 74% specificity and 96% sensitivity in a meta-analysis of 3202 CRC patients and 7284 controls [105,106,107]. Subsequent studies in younger populations reported 96.3% specificity and 90.8% sensitivity, but limited sample size (27 CRC patients, 87 controls) constrains broader applicability [108]. Beyond SEPT9, the SpecColon test targets SFRP2 and SDC2 methylation, showing promise in identifying CRC-associated epigenetic changes [109]. Other potential markers include IKZF1, BCAT1, NTRK3, SRFP1-2, EHD3, TMEM240, SMAD3, and VIM, though further validation is necessary to confirm clinical utility and integration into routine CRC screening workflows [142,143].

The available data show that approximately three out of four FIT+ patients are negative for diagnostic endoscopic evaluation. A recent Italian study evaluated the role of cfDNA in identifying FIT+ patients at higher risk of having a positive colonoscopy. This trial enrolled 711 FIT+ patients, who subsequently underwent both the QuantiDNA™ (Pleasanton, CA, USA) test and colonoscopy. It was found that the QuantiDNA™ (Pleasanton, CA, USA) test is effective in reducing the need for colonoscopies by approximately 33.4%, proving to be non-inferior to the standard of care in the early identification of colorectal lesions [110].

### 2.2. Non-Coding RNAs

Recent studies highlight the potential of miRNAs as non-invasive biomarkers for CRC due to their dual role as oncogenes and tumor suppressors in processes like proliferation, apoptosis, angiogenesis, and metastasis [111,144]. Synthesized in the nucleus and present in extracellular fluids like blood and serum, miRNAs are released via secretory mechanisms, mainly in extracellular vesicles (exosomes) or passively from damaged cells [112]. Their stability in bodily fluids enhances their diagnostic and prognostic value, enabling early cancer detection and monitoring [111,112]. Techniques such as qRT-PCR, microarray analysis, and NGS facilitate the identification of tumor-specific miRNA expression profiles relevant to CRC diagnosis [101].

The growing interest in miRNAs as early CRC biomarkers has led to studies on miR-21, which showed promising diagnostic accuracy with 77% sensitivity and 83% specificity [113,114,115,116]. Other miRNAs, including miR-210, miR-144-3p, and miR-1246, have also been identified as potential CRC biomarkers, though further validation is needed [115,117,118,119,120].

Research indicates that miRNA panels combining multiple markers often outperform single miRNAs in diagnostic accuracy. For instance, a panel comprising miR-193a-5p, miR-210, miR-513a-5p, and miR-628-3p achieved an AUC of 0.92, with 90% sensitivity and 80% specificity [118]. However, due to the heterogeneous expression of miRNAs, single-marker approaches are less reliable, and comprehensive panels may more accurately reflect tumor-specific profiles [121,145].

LncRNAs, detectable in blood, plasma, serum, and urine, exhibit significant diagnostic potential due to their stability in bodily fluids and resistance to RNase degradation [122,123,124,125]. They can circulate within apoptotic bodies, microvesicles, and exosomes, facilitating their transport and protecting them from degradation [126]. Aberrant lncRNA expression can influence key oncogenic pathways such as WNT/β-catenin, PI3K/Akt, EGFR, NOTCH, mTOR, and TP53, linking them to CRC progression [127]. Several circulating lncRNAs have been investigated as potential CRC biomarkers. Notably, CCAT1 and HOTAIR are consistently reported as elevated in CRC patients compared to healthy controls, suggesting their potential role in early diagnosis [128]. Additionally, other lncRNAs such as BLACAT1, CRNDE, NEAT1, and UCA1 have also demonstrated diagnostic potential, with some identified within exosomes, indicating a role in intercellular communication and metastasis [146,147]. Exosomal forms of 91H, CRNDE-h, UCA1, TUG1, and multiple LNCV6 transcripts (e.g., LNCV6_116109, LNCV6_98390) are under investigation as potential components of multi-marker panels for improved diagnostic accuracy [146,147].

CircRNAs, characterized by stable covalently closed-loop structures, exhibit resistance to exonuclease degradation [129]. They play roles in gene transcription and splicing regulation and are gaining attention as potential biomarkers for early CRC detection. Exosomal circ_0004771, significantly upregulated in CRC patients and reduced post surgery, suggests potential for early diagnosis and monitoring [130]. Although circRNA research is still emerging, their structural stability positions them as promising diagnostic and prognostic markers. Currently, the only FDA-approved lncRNA-based test, PCA3, is available for prostate cancer, highlighting the potential for similar circRNA-based diagnostics in CRC [131].

### 2.3. Other Biomarkers

Insulin-like growth factors (IGF-1 and IGF-2) promote tumor growth and act as mitogens for colon mucosa, advancing cancer progression [132]. Insulin-like growth factor-binding protein 2 (IGFBP-2), which binds IGF-2, is linked to cancer metastasis through its interaction with HSP27 [133]. Elevated serum IGFBP-2 levels in CRC patients correlate with neoplastic changes and increased CEA concentrations, indicating potential for early diagnosis and monitoring, though its standalone sensitivity and specificity are limited [134].

Pyruvate Kinase M2 (PKM2), a glycolytic enzyme, is overexpressed in CRC, gastric cancer, and adenomas [148]. While it shows high sensitivity as a blood and fecal biomarker, its specificity remains suboptimal when used alone [149].

Dickkopf-3 (DKK3), part of the tumor endothelial marker family, is frequently silenced in CRC due to promoter hypermethylation [150]. Although its role is not fully defined, DKK3 is implicated in angiogenesis and tumor progression, suggesting its potential as a CRC biomarker [151,152].

Individually, IGFBP-2, PKM2, and DKK3 exhibit limited diagnostic accuracy, but a combined panel has demonstrated improved performance, with 57% sensitivity for early-stage CRC, 76% for advanced stages, and 95% specificity, underscoring its potential as a non-invasive blood-based diagnostic tool [134].

## 3. Localized CRC

Following surgical treatment, adjuvant chemotherapy is recommended for patients with stage II-III CRC. However, the existing risk stratification factors primarily rely on clinical and pathological parameters, which, though important, do not fully encompass all the nuances necessary for optimal patient management. Consequently, the possibility of overtreatment or undertreatment in patients with early-stage CRC remains an ongoing concern. Therefore, the identification of supplementary diagnostic tools to refine patient selection and guide subsequent therapeutic decisions remains an unmet clinical need. From this perspective, identifying biomarkers that can detect MRD represents an important goal [99,153,154].

### 3.1. Detection of MRD in CRC: Plasma-Only vs. Tumor-Informed Assay

The detection of MRD in CRC is a rapidly evolving field with significant implications for post-surgical management and adjuvant treatment decision-making. MRD refers to residual tumor cells that persist after primary treatment and may lead to recurrence. Liquid biopsy, particularly through ctDNA analysis, has emerged as a promising non-invasive method for detecting MRD, providing molecular-level insights not achievable through conventional imaging [155,156,157,158].

Two main methodologies exist for ctDNA-based MRD detection: plasma-only and tumor-informed assays. Plasma-only assays, also known as tumor-agnostic assays, analyze cfDNA from blood samples without prior tissue sequencing [155]. These assays target predefined gene panels commonly mutated in CRC, such as KRAS, TP53, and APC [159,160,161]. Examples of commercial plasma-only assays include Guardant Reveal™ (Redwood City, CA, USA) and AVENIO ctDNA Surveillance Kit V2™, which utilize targeted sequencing and fragmentomics to enhance MRD detection [162,163]. Fragmentomics leverages differences in fragment size and integrity to distinguish ctDNA from normal cfDNA, improving the detection of low-frequency mutations [164,165].

While plasma-only assays offer a simpler workflow without tumor sampling, they may miss mutations specific to a patient’s tumor, increasing the risk of false negatives. Additionally, their sensitivity can be affected by ctDNA shedding rates, which vary by tumor size, location, and biological characteristics, particularly in tumors with low proliferative capacity or poor vascularization [158,160,165].

In contrast, tumor-informed assays adopt a personalized approach by first sequencing the primary tumor to identify patient-specific mutations [166,167]. These mutations are then targeted in subsequent cfDNA samples, allowing more sensitive MRD detection, especially in cases with low ctDNA shedding [155,166,168]. However, these assays require initial tumor tissue sampling, which may not always be feasible, and may miss new mutations arising after the biopsy [158,166].

The choice between plasma-only and tumor-informed assays depends on factors such as tumor biology, disease stage, and clinical context. Plasma-only assays may suit early-stage CRC with high ctDNA shedding, while tumor-informed assays offer greater sensitivity in advanced or low-shedding cases by targeting patient-specific mutations [157,158,165]. The timing of blood collection is crucial for MRD detection accuracy, as ctDNA levels can fluctuate in the first weeks after surgery [169,170]. Blood samples are typically collected five to eight weeks after surgery to minimize the detection of DNA released due to surgical trauma rather than residual disease [169,170,171]. Testing before initiating adjuvant chemotherapy can also provide a baseline for treatment response and recurrence monitoring.

Despite their potential, both plasma-only and tumor-informed assays face challenges, including low ctDNA concentration, high cfDNA background, and false positives due to clonal hematopoiesis. Advances in NGS, fragmentomics, and epigenetic analysis may further refine MRD detection, enhancing both sensitivity and specificity. Integrating multi-omic approaches could provide a more comprehensive assessment of residual disease, supporting personalized treatment strategies in CRC [155,172,173].

### 3.2. Patient Selection, Treatment Modulation, and Follow-Up in Non-Metastatic Colon Cancer

Several studies have evaluated the prognostic role of ctDNA as a marker of MRD in CRC, aiming to guide adjuvant therapy (Table 3).

Among these, a prospective study by Reinert et al., involving 130 patients with stage I–III CRC, detected preoperative ctDNA in 88.5% of cases [174]. Post treatment, ctDNA analysis identified 87.5% of relapses. On postoperative day 30, ctDNA-positive patients had a 7-fold increased risk of relapse (HR: 7.2; 95% CI: 2.7–19.0; *p* < 0.001), which rose to 17-fold following adjuvant chemotherapy (HR: 17.5; 95% CI: 5.4–56.5; *p* < 0.001). Post-treatment surveillance revealed that ctDNA-positive patients had a 43.5-fold increased recurrence risk (HR: 43.5; 95% CI: 9.8–193.5; *p* < 0.001). During post-treatment surveillance, patients who tested positive for ctDNA were over 40 times more likely to experience disease recurrence than ctDNA-negative patients (HR: 43.5; 95% CI: 9.8–193.5; *p* < 0.001). In multivariate analyses, ctDNA status remained an independent predictor of relapse, even after adjusting for established clinicopathologic risk factors. Serial ctDNA assessments detected recurrences up to 16.5 months earlier than radiologic imaging, with a mean lead time of 8.7 months [174].

The CIRCULATE-Japan project, comprising the observational GALAXY study and phase III VEGA and ALTAIR trials, assessed ctDNA-guided therapy in stages II–IV of CRC [175,176]. A 2023 analysis involving 2280 patients showed that MRD-positive (MRD+) patients had worse disease-free survival (DFS) at 24 months (HR: 16.9, *p* < 0.0001) and benefited from adjuvant chemotherapy (HR: 0.4, *p* < 0.0001). Among those receiving 6 months of adjuvant chemotherapy, DFS significantly improved (HR: 0.4, *p* = 0.002) [177]. An update presented at the ASCO 2024 Annual Meeting revealed that ctDNA-positive patients who achieved clearance had a significantly lower relapse risk (HR: 5.4; 95% CI: 3.58–7.67; *p* < 0.0001). Among 181 ctDNA+ patients treated with adjuvant therapy, 72.9% achieved ctDNA clearance, with 54% maintaining clearance and 46% experiencing re-emergence. Notably, patients with sustained clearance demonstrated markedly better outcomes (HR: 32.57; 95% CI: 9.94–106.76; *p* < 0.0001) [178].

The DYNAMIC II trial assessed ctDNA-guided therapy in stage II CRC, enrolling 455 patients randomized 2:1 to ctDNA-guided treatment or standard care (SOC). Only 15% of patients in the ctDNA arm received adjuvant chemotherapy compared to 28% in the SOC group, without compromising 3-year relapse-free survival (3 y RFS: 91.7% vs. 92.5%) [179]. However, chemotherapy regimens differed, with oxaliplatin-based therapy predominating in the ctDNA group and single-agent fluoropyrimidine in the control group, representing a study limitation. Another limitation was that ctDNA-negative patients with resected T4 tumor had recurrence rate similar to ctDNA-positive patients, suggesting that ctDNA alone may not be sufficient to assess recurrence risk in stage II colon cancer [179].

At the ASCO 2025 Annual Meeting, the DYNAMIC-III trial confirmed the prognostic value of postoperative ctDNA in stage III colon cancer. Patients with detectable ctDNA showed a high risk of recurrence. However, ctDNA-guided escalation of adjuvant chemotherapy, including FOLFOXIRI, did not improve RFS, underscoring the urgent need for alternative therapeutic strategies in this high-risk subgroup [180].

The PEGASUS trial assessed ctDNA-guided treatment adjustments in stage III and T4N0 stage II CRC patients. Preliminary results showed that 7% of ctDNA-negative patients relapsed after treatment de-escalation, mainly in the peritoneum and lungs, associated with low ctDNA shedding. Among ctDNA-positive patients treated with CAPOX and subsequently switched to FOLFIRI, 46% achieved ctDNA clearance. These findings offer encouraging early evidence for the potential of ctDNA-guided therapeutic modifications [181].

The COBRA trial focused on ctDNA clearance after adjuvant chemotherapy in the ctDNA+ cohort, but phase II was terminated early due to a high rate of false positives in interim analysis [182].

At the ASCO Gastrointestinal Cancers Symposium 2025, results from the CALGB/SWOG 80702 trial confirmed ctDNA as a strong prognostic marker in stage III colon cancer and suggested its potential as a predictive biomarker that would benefit from adding celecoxib to adjuvant FOLFOX chemotherapy [183].

Ongoing trials, including CIRCULATE-US, AFFORD, DYNAMIC-III, and CLAUDIA, are investigating intensified chemotherapy regimens like FOLFOXIRI in MRD-positive patients (NCT05174169, NCT05427669, ACTRN126170015, NCT05534087). Concurrently, phase II trials FANTASTIC and AURORA are exploring FOLFOXIRI and FOLFOXIRI plus bevacizumab in resected oligometastatic CRC [184,185]. Additionally, several studies aim to validate ctDNA for monitoring recurrence and identifying candidates for further intervention in post-adjuvant therapy [186,187,188,189,190].

### 3.3. Liquid Biopsy in Non-Metastatic Rectal Cancer

Preoperative chemoradiotherapy followed by total mesorectal excision (TME) is the standard treatment for non-metastatic rectal adenocarcinoma [189,191]. Recently, total neoadjuvant treatment (TNT) has shown superiority in pathological complete response (pCR) and overall survival (OS), as demonstrated in the PRODIGE 23 trial [192,193,194]. In this context, ctDNA may serve as a predictive marker for treatment response and guide postoperative therapy adjustments (Table 4).

Appelt et al. reported that ctDNA-positive patients had worse outcomes in terms of OS compared to ctDNA-negative patients [195]. In a prospective study by Murahashi et al., ctDNA was detected in 57.6% of baseline samples and 22.3% of samples post treatment in 85 patients with locally advanced rectal cancer (LARC). ctDNA clearance from baseline to post-neoadjuvant treatment was associated with higher pCR rates (*p* = 0.0276). Postoperative ctDNA positivity was linked to worse RFS, especially when combined with CEA analysis (ctDNA, *p* = 0.0127; CEA, *p* = 0.0105) [196].

A meta-analysis by Chang et al. further supported these findings, showing that post-neoadjuvant ctDNA positivity was associated with worse RFS (HR: 9.16; 95% CI: 5.48–15.32), OS (HR: 8.49; 95% CI: 2.20–32.72), and lower pCR rates (OR: 0.40; 95% CI: 0.18–0.89) [197]. Additional studies confirmed ctDNA as a predictive and prognostic marker in this setting [198,199,200].

Tie et al. also demonstrated a significant association between ctDNA positivity, post-neoadjuvant therapy, surgery, and worse RFS (HR: 6.6 and 13.0, respectively) [201].

Building on these data, the ongoing DYNAMIC RECTAL trial is assessing the role of ctDNA in guiding adjuvant chemotherapy decisions in LARC. Patients are randomized to SOC or ctDNA-guided chemotherapy, with adjuvant chemotherapy use as the primary endpoint and 3-year RFS as a key secondary endpoint. Preliminary ASCO 2024 data from 230 LARC patients showed that 46% of the ctDNA-guided group received adjuvant chemotherapy, compared to 77% in the control group. The 3-year RFS was 76% in the ctDNA-guided group versus 82% in the control group, suggesting potential for reducing overtreatment. Additionally, among ctDNA-negative patients, lung-only recurrences accounted for 80% of relapses, indicating a potential limitation in detecting pulmonary metastases [202]. However, the small sample size and lack of TNT inclusion limit definitive conclusions regarding the non-inferiority of the ctDNA-guided approach.

## 4. Advanced Disease: Target and Response Evaluation in Metastatic Colorectal Cancer

The use of liquid biopsy in metastatic CRC (mCRC) has transformed the clinical management of this disease, providing a non-invasive means of continuously monitoring tumor dynamics. The decision-making process for first-line treatment in mCRC is highly dependent on the molecular characterization of the tumor. Key biomarkers, including RAS, BRAF, HER2, and mismatch repair/microsatellite instability (MMR/MSI) status, are routinely assessed to guide therapeutic strategies [203]. Traditionally, molecular profiling is conducted through tissue biopsy, but liquid biopsy has emerged as a crucial complementary tool, particularly when tissue samples are unavailable or inadequate. ctDNA, CTCs, exosomal RNA, and extracellular vesicle DNA (evDNA) offer real-time molecular insights, addressing the limitations of spatial and temporal tumor heterogeneity [204].

### 4.1. Molecular Concordance, Dynamic Profiling, and Prognostic Insights

The primary challenge in integrating liquid biopsy into clinical practice is ensuring its reliability and concordance with tissue-based molecular analyses. Several studies report high concordance rates for RAS and BRAF mutations between liquid and tissue biopsies (80–100%), particularly when NGS and ddPCR techniques are used [205,206,207,208,209]. However, ctDNA detectability and concordance can be influenced by factors such as tumor burden and metastatic site. Patients with liver-dominant disease often show higher ctDNA levels, while those with lung or peritoneal metastases exhibit lower ctDNA shedding, resulting in reduced sensitivity (Se) and specificity (Sp), sometimes as low as 60% [206,210,211,212,213,214].

Chemotherapy prior to ctDNA extraction can lower the variant allele fraction (VAF): higher VAFs are noted in liver metastases and lower in cancer with mucinous differentiation, contributing to discordance [210,213,214]. Additionally, a long interval between tissue and blood sample collection may reduce concordance due to tumor evolution and emerging genetic alterations.

Unlike single-timepoint tissue biopsies, liquid biopsy can capture dynamic molecular changes over time. Some mCRC patients initially classified as RAS-mutant in tissue may later present as RAS wild-type (wt) in ctDNA, a phenomenon known as “RAS conversion” [215,216,217]. Pesola et al. reported RAS conversion rates ranging from 5.5% to 78%, influenced by methodological differences [218]. This shift, potentially due to chemotherapy-induced clonal selection, may allow patients to regain sensitivity to anti-EGFR therapies, underscoring the value of serial ctDNA monitoring for real-time molecular reassessment [219,220,221,222,223,224,225,226,227,228].

High baseline ctDNA levels have been strongly associated with worse prognosis, a trend that persists even in later treatment lines [229,230,231]. Elevated RAS VAF at baseline correlates with poor prognosis, even in cases where tissue biopsy indicates RAS-wt status [232,233,234]. Conversely, patients with RAS mutations in tissue but not in plasma tend to have better survival outcomes [233].

Monitoring ctDNA during treatment can provide early indications of therapeutic response, often detectable as early as two weeks after chemotherapy initiation, much earlier than radiological assessments [230,235,236,237]. Lower ctDNA levels after induction are linked to longer PFS, with ctDNA dynamics reflecting those of traditional tumor markers like CEA [230,238,239,240].

While ctDNA is the most widely used plasma biomarker, CTCs provide comprehensive tumor information and can serve as a platform for cell culture. CTC presence at CRC diagnosis is associated with poor prognosis, especially when expressing mesenchymal markers like vimentin, linked to metastatic disease and decreased DFS [241,242,243,244].

Tumor-derived extracellular vesicles (EVs) also offer a promising source of circulating DNA (evDNA), carrying driver mutations with acceptable sensitivity and high specificity [245,246]. Among epigenetic markers, ctDNA hypermethylation, such as NPY promoter methylation detected via ddPCR, is gaining interest. Higher baseline methylation correlates with poor prognosis, and its decline during treatment mirrors radiological response [247,248].

Exosomes, present in most body fluids, carry diverse nucleic acids, including miRNA, lncRNA, and circRNA [249]. These RNAs are not only useful in distinguishing metastatic from non-metastatic CRC but also hold potential in detecting chemoresistance, as demonstrated for miRNAs [250,251]. Predictive models are being developed, incorporating specific lncRNAs and circRNAs [252,253,254,255,256].

### 4.2. Key Molecular and Immunohistochemical Targets in mCRC Identifiable Through Liquid Biopsy

#### 4.2.1. Anti-EGFR Therapy: Treatment Selection, Resistance Mechanisms, and Rechallenge Strategies

In RAS-wt, mismatch repair-proficient/microsatellite stable (pMMR/MSS) mCRC, first-line treatment typically involves chemotherapy (fluoropyrimidine plus oxaliplatin or irinotecan) combined with epidermal growth factor receptor inhibitors (EGFRis) like cetuximab or panitumumab [203]. Due to primary resistance to EGFRi conferred by RAS mutations, comprehensive molecular testing is essential prior to treatment initiation.

Liquid biopsy enhances RAS mutation detection by overcoming the spatial and temporal limitations of tissue biopsy. Several studies have shown that ctDNA analysis can more accurately identify RAS mutations in patients initially considered RAS wt [212,234,237,257]. Table 5 summarizes the main studies evaluating the role of liquid biopsy in treatment selection and anti-EGFR rechallenge in mCRC.

The LIBImAb study presented at ASCO 2023 reported RAS mutations in 9.5% of patients initially deemed RAS-wt by tissue biopsy [258], while the FIRE-4 trial found a 13% discordance, underscoring the importance of baseline ctDNA testing to refine EGFRi therapy [259,260].

Beyond RAS, liquid biopsy enables broader molecular profiling, identifying resistance mechanisms involving ERBB2, EGFR, FGFR1, PIK3CA, PTEN, AKT1, MAP2K1, and gene amplifications in KRAS, ERBB2, MET, and fusions in ALK, ROS1, NTRK1-3, and RET [275,276,277]. The VALENTINO trial identified these alterations as predictors of poor PFS and OS during panitumumab maintenance [261]. Real-time ctDNA monitoring is key to capturing emerging resistance mutations, particularly secondary RAS mutations often missed by tissue biopsy [234,257,278,279]. However, chemotherapy plus EGFRi appears to reduce resistance mechanisms involving ERBB2, BRAF, PIK3CA, MET, FLT3, and MAP2K1 [280,281,282,283,284,285,286].

Adaptive or switch maintenance therapy is being investigated in patients developing resistance mutations during first-line EGFRi treatment, through trials such as MODUL (NCT02291289), Rapid 1 (NCT04786600), and MoLiMoR (NCT04554836). The COPERNIC trial (NCT05487248) is assessing ctDNA dynamics to guide third-line therapy. It features an initial pilot phase to define optimal tumor fraction (TF) thresholds and timing, followed by a randomized phase to evaluate the clinical impact of ctDNA-driven treatment adjustments.

Rechallenge with EGFR inhibitors is also being explored based on the hypothesis that resistant clones may regress after a drug-free interval, potentially restoring drug sensitivity [286,287]. Early rechallenge strategies relied solely on baseline tissue RAS-wt status, disregarding molecular evolution over time [288,289].

In the phase II CRICKET trial, cetuximab plus irinotecan met its objective response rate (ORR) endpoint; however, ctDNA analysis revealed that nearly 50% of patients harbored RAS mutations at rechallenge, limiting responses to RAS-wt cases [262]. The Japanese REMARRY trial refined this approach by requiring ctDNA-confirmed RAS-wt status before rechallenge with panitumumab plus irinotecan. Although the ORR endpoint was not met, longer EGFRi-free intervals were associated with better outcomes [263,264].

The VELO trial showed that panitumumab plus trifluridine/tipiracil provided prolonged benefit in patients with ctDNA-confirmed RAS/BRAF-wt [265]. The CAVE study demonstrated favorable OS with cetuximab plus avelumab, although patient selection was not ctDNA-driven [266].

The CHRONOS trial confirmed the utility of ctDNA-based ultra-selection for panitumumab rechallenge [267]. Similarly, the REMARRY-PURSUIT trial evaluated ctDNA RAS dynamics to guide anti-EGFR rechallenge, reporting a modest ORR of 14% due to the exclusion of BRAF- and EGFR-mutated cases [263].

Mariani et al. highlighted the importance of the EGFRi-free interval, showing that rechallenge after ≥14–16 months was associated with better PFS and OS, supporting the use of both clinical and molecular criteria for patient selection [268].

A recent preplanned baseline translational analysis from the randomized phase II CAVE-2 GOIM trial, avelumab plus cetuximab versus cetuximab alone as anti-EGFR rechallenge in refractory ctDNA RAS/BRAFV600E wt mCRC patients, showed that baseline ctDNA CGP (FoundationOne Liquid CDx) identifies acquired RAS/BRAFV600 and non-RAS/BRAFV600 resistance alterations, supporting liquid-biopsy–guided hyper-selection for anti-EGFR rechallenge in refractory RAS/BRAFV600E–wild-type mCRC [269].

Moreover, the PARERE phase II trial, presented at ESMO GI 2025, evaluated the optimal sequencing of panitumumab and regorafenib in RAS/BRAF-wt mCRC patients selected by ctDNA. Administering panitumumab before regorafenib resulted in improved outcomes, supporting ctDNA-guided rechallenge strategies in chemorefractory settings [272].

Novel strategies to overcome resistance are under investigation. For example, trametinib plus panitumumab was tested in patients with ctDNA-detected RAS, BRAF, or MAP2K1 mutations but was discontinued due to poor tolerability [270]. Similarly, targeting MET amplification with tepotinib plus cetuximab showed promise but faced accrual limitations (NCT04515394) [271].

Interestingly, early studies suggest that VEGF inhibitors (VEGFis) may promote reversion to RAS-wt clones in RAS-mut patients, potentially restoring sensitivity to EGFR inhibitors [290], a hypothesis currently under further investigation.

Efforts are also underway to target actionable mutations beyond RAS. The C-PRECISE-01 trial is selecting patients with ct-DNA detected PIK3CA mutations for treatment with cetuximab plus the PI3K inhibitor MEN1611 [273]. Meanwhile, the OrigAMI-1 trial showed that ctDNA-guided rechallenge with amivantamab in left-sided RAS/BRAF/EGFR-wt mCRC patients resulted in greater clinical benefit when administered after a longer EGFRi-free interval (>8.8 months), highlighting the importance of timing and molecular selection in anti-EGFR re-treatment [274].

Rechallenge strategies are increasingly reliant on ctDNA-based assessment of RAS mutational status. Trials such as PARERE (NCT04787341) [272], CAVE-2 (NCT05291156) [269], CITRIC [291], PULSE (NCT03992456), and CAPRI 2 GOIM (NCT05312398) incorporate NGS-based ctDNA profiling to guide treatment across various lines.

#### 4.2.2. Targeting BRAF-Mutant: Applications and Overcoming Resistance

BRAF-V600E mutations, detected in approximately 10% of mCRC cases, are associated with poor prognosis and resistance to standard chemotherapy [292,293]. Liquid biopsy provides a sensitive, non-invasive method for detecting this alteration, particularly in patients with liver metastases. Recent advancements in NGS have achieved detection rates exceeding 90%, outperforming conventional tissue biopsy [206,212,213,294,295,296].

In BRAF-V600E-mutant mCRC, treatment after progression on standard chemotherapy plus bevacizumab, or after immune checkpoint inhibitors in dMMR/MSI-H tumors, involves dual inhibition of the BRAF and EGFR pathways. The encorafenib–cetuximab regimen is the current standard in this setting [297,298,299,300]. Investigational strategies are exploring BRAF-targeted therapy in earlier lines, either as monotherapy or combined with chemotherapy or immunotherapy for dMMR/MSI-H cases [295,301,302].

Despite these efforts, response rates remain modest, with ORR around 20% and median duration of response under six months. Primary resistance remains a challenge [298]. Identifying resistance mechanisms, both primary and acquired, is crucial to refine patient selection and optimize combination strategies. Liquid biopsy enables real-time monitoring of clonal evolution and emerging resistance [303,304]. Table 6 provides an overview of studies assessing the impact of ctDNA monitoring on treatment outcomes in BRAF-mutant mCRC.

The BEACON trial provided key insights into resistance mechanisms. ctDNA analysis at baseline and progression confirmed BRAF-V600E in nearly 90% of cases and identified acquired alterations in KRAS, NRAS, MAP2K1, and MET amplification [305]. These alterations were observed at similar rates in both the BRAF-EGFR doublet and BRAF-EGFR-MEK triplet arms, with the triplet arm showing slightly fewer MAPK mutations but more pathway-related amplifications. At least one resistance-associated alteration was identified in over 60% of progressing patients, often with multiple concurrent mutations [308,309].

Alterations in DNA repair and PI3K signaling genes also emerged as key resistance mechanisms, with baseline mutations linked to early progression and poor outcomes [308]. RNF43 mutations, particularly in MSS tumors, were associated with better responses to BRAF-targeted therapy, potentially due to WNT pathway inhibition mitigating MAPK-driven progression [305,308,309,310].

Efforts to overcome resistance have led to small-scale rechallenge studies. Ji et al. reported three cases where rechallenge strategies led to responses, including one patient with a rising BRAF-V600E variant allele fraction (VAF) but no new MAPK mutations who achieved an eight-cycle response to encorafenib–cetuximab [306]. Another patient with prior MET amplification responded to vemurafenib–cetuximab–irinotecan following MET amplification decrease post-intervening therapy [306]. Oddo et al. also reported a rechallenge using vemurafenib and crizotinib in a patient with BRAF-V600E and MET amplification, with ctDNA dynamics predicting radiologic progression by eight weeks [307].

Plasma BRAF-V600E VAF has emerged as a key surrogate for tumor burden and prognosis. High baseline VAF (≥2%) correlates with increased metastatic load, liver involvement, and poorer clinical outcomes but also predicts greater benefit from triplet therapy including MEK inhibition [304,309]. In the BEACON trial, lower VAF was associated with longer OS across all treatment arms [293,306]. Ye et al. confirmed that patients with low or undetectable baseline VAF (<5%) had superior PFS and OS, with a >50% decline in VAF during treatment correlating with partial response or disease stabilization [308].

#### 4.2.3. Emerging Biomarkers: HER2, MSI, Exosomal RNA, and Methylation Analysis

The detection of additional driver alterations through liquid biopsy remains underexplored, but the growing use of NGS assays has enhanced the capacity to assess multiple genetic alterations simultaneously, offering a more comprehensive view of tumor heterogeneity. Despite this, concordance between plasma-based and tissue-based analyses varies depending on the specific alteration.

HER2/ERBB2 amplification, typically assessed by immunohistochemistry (IHC) or in situ hybridization (ISH), shows variable concordance with plasma-based copy number variation (CNV) analysis, with agreement rates ranging from 66.7% to 97.9% [311,312]. Nevertheless, ctDNA-based CNV has proven predictive of response to HER2-targeted therapies [313]. Siravegna et al. established a CNV threshold of ≥2.4 copies adjusted for ctDNA fraction, correlating with PFS and objective response in HER2-amplified patients identified by tissue analysis. The DESTINY-CRC01 trial similarly reported that higher HER2 CNV or serum HER2 extracellular domain levels predicted better responses to trastuzumab deruxtecan [314].

ctDNA profiling has also demonstrated predictive value in assessing response to trastuzumab and pertuzumab, with similar accuracy when compared to tissue analysis [315]. A decline in ctDNA levels within three weeks of treatment initiation was associated with favorable outcomes. Meanwhile, ctDNA identified receptor tyrosine kinase (RTK), RAS, or PI3K pathway mutations in 67% of non-responders, compared to 19% in matched tissue samples, underscoring the utility of liquid biopsy in detecting resistance mechanisms.

ERBB2 mutations, primarily linked to resistance to EGFRi, may also emerge during HER2-targeted therapy, though their predictive role remains unclear [315,316,317]. Ongoing phase II studies are investigating this phenomenon [317].

In dMMR/MSI-H mCRC, ctDNA-based MSI detection using ddPCR or NGS demonstrated high accuracy [318,319,320], suggesting its use in clinical practice, particularly when tissue samples are unavailable or when discordant results occur between MSI status and clinical features [321]. Despite lower tumor heterogeneity in this subgroup, discordance between tissue and plasma remains a concern [322,323,324,325].

Moreover, ctDNA enables dynamic monitoring of tumor mutational burden (TMB) and MSI, providing insights into treatment response and resistance in patients receiving immune checkpoint inhibitors (ICIs) [326,327]. This is especially relevant as up to 30% of dMMR/MSI-H patients exhibit primary resistance to immunotherapy [328]. In such cases, liquid biopsy may help identify non-responders earlier, potentially guiding timely treatment adjustments, such as switching to dual checkpoint blockade or ICI–chemotherapy combinations.

A case series by Kasi et al. described three dMMR/MSI-H mCRC patients who progressed on pembrolizumab but responded to nivolumab plus ipilimumab, with ctDNA becoming undetectable shortly after dual ICI therapy initiation [329].

In MSS mCRC, ctDNA has provided insights into immunotherapy mechanisms. In the AVETUX trial, which evaluated FOLFOX plus cetuximab and avelumab in RAS/BRAFwt MSS patients, early ctDNA reduction at four weeks correlated with high response rates. Additionally, the clonality and diversity of peripheral and tumor-infiltrating lymphocytes after three treatment cycles predicted disease control [330,331].

Interestingly, PD-L1 mutations emerged in a subset of responders, likely driven by selective pressure from avelumab, resulting in reduced binding affinity but enhanced T cell-mediated killing [332]. These findings illustrate the potential of liquid biopsy to monitor adaptive changes during immunotherapy and refine biomarker-driven strategies in mCRC.

## 5. Conclusions

Liquid biopsy has emerged as a promising non-invasive approach for detecting and monitoring CRC, offering molecular insights through ctDNA, CTCs, ncRNAs, and other biomarkers. While its clinical utility is well-established in metastatic CRC, its application in non-metastatic disease, particularly for screening and MRD detection, remains in the exploratory phase.

Based on current evidence, we propose a stage-specific conceptual algorithm for integrating liquid biopsy into CRC management, encompassing screening, MRD detection in localized disease, and real-time monitoring in metastatic settings (Figure 2). This flowchart summarizes the most promising applications reported in the literature and may serve as a reference for future clinical validation.

In metastatic CRC, liquid biopsy supports real-time treatment monitoring, resistance detection, and tracking of tumor evolution. Technologies such as NGS, dPCR, and methylation-based assays enable the identification of actionable alterations (e.g., RAS, BRAF, HER2), facilitating targeted therapy selection and rechallenge decisions. ctDNA profiling also captures emerging resistance mechanisms, supporting timely therapeutic adaptation. However, implementation is challenged by variability in assay sensitivity, particularly in low-shedding tumors, as well as occasional discordance between tissue and plasma-based results.

In non-metastatic CRC, the role of liquid biopsy is less defined but gaining traction, particularly for MRD detection and adjuvant therapy tailoring. Trials like DYNAMIC II and CIRCULATE-Japan have demonstrated its potential to identify patients at high risk of relapse and to spare low-risk individuals from unnecessary chemotherapy. Nonetheless, most studies to date are single-arm, with limited follow-up and lacking randomized validation. Technical issues, including inconsistent ctDNA shedding and sampling intervals, as well as confounding from clonal hematopoiesis and tumor heterogeneity, further complicate interpretation, and clinical adoption.

Across all stages, the lack of standardized methodologies limits cross-study comparability and slows the development of uniform clinical recommendations. Many rechallenge trials in metastatic CRC, despite being prospective, are constrained by small sample sizes, heterogeneous inclusion criteria (e.g., EGFRi-free interval, molecular profiling strategies), and variable ctDNA platforms.

Despite these limitations, the integration of molecular profiling, adaptive trial designs, and ctDNA monitoring marks a clear shift toward personalized treatment. To fully realize the potential of liquid biopsy in clinical practice, several hurdles must be addressed: the development of analytically and clinically validated assays with high sensitivity and specificity; consensus on sampling intervals and result interpretation; and robust cost-effectiveness analyses. Expanding biomarker panels to include additional resistance pathways and adopting multi-omic frameworks, combining ctDNA with transcriptomic and proteomic data, may further enhance patient stratification and therapy personalization.

Harmonization of protocols for serial sampling, timing, and thresholds is essential to improve reproducibility and facilitate meaningful comparisons across studies. Large, prospective randomized trials using standardized ctDNA methodologies remain the cornerstone for establishing definitive clinical utility.

In parallel, practical and ethical aspects must be addressed. Cost-effectiveness will determine feasibility in resource-limited settings, while inequitable access to advanced diagnostics may widen disparities in care. Additionally, incidental findings, such as germline mutations, raise ethical concerns around patient counseling and data interpretation. Addressing these challenges is critical to ensuring responsible and equitable adoption.

From a regulatory perspective, liquid biopsy is gaining recognition. The U.S. FDA has granted Breakthrough Device designation to multiple ctDNA assays (e.g., Signatera™ by Natera, Austin, TX, USA) for MRD detection in CRC and other cancers [333]. Moreover, both the NCCN and ESMO have begun incorporating ctDNA into clinical guidelines for applications such as molecular profiling in metastatic CRC and MRD-driven therapy decisions [334,335]. However, broader clinical integration depends on the generation of high-quality prospective data and methodological standardization.

In conclusion, liquid biopsy holds great promise as a minimally invasive, dynamic tool across all CRC stages. Overcoming current limitations will be key to making it a cornerstone of precision oncology, enabling more refined, data-driven treatment decisions, and ultimately improving patient outcomes.

## Figures and Tables

**Figure 1 ijms-26-07619-f001:**
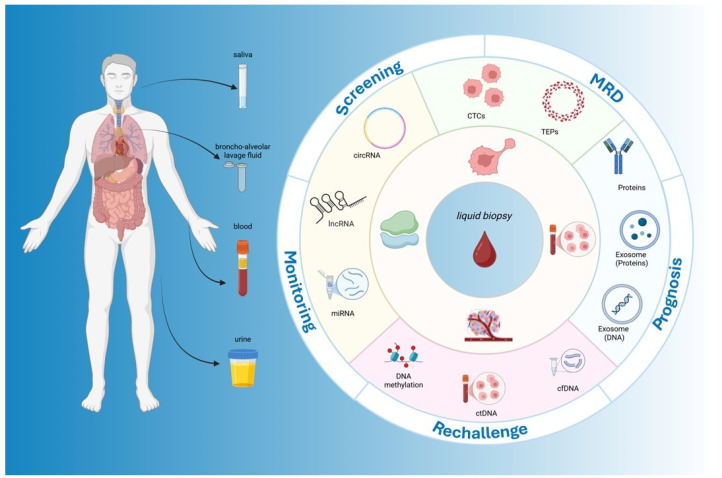
Clinical applications of liquid biopsy in colorectal cancer. The figure illustrates the sources, analytes, and clinical applications of liquid biopsy in CRC. On the left, different body fluids (blood, urine, saliva, and bronchoalveolar lavage fluid) are depicted as collection matrices. The central circle highlights the key analytes, including circulating tumor DNA (ctDNA), cell-free DNA (cfDNA), microRNAs (miRNAs), long non-coding RNAs (lncRNAs), circular RNAs (circRNAs), proteins, tumor-educated platelets (TEPs), and exosomes. The outer ring summarizes the major clinical applications across the CRC continuum: screening, minimal residual disease (MRD) detection, prognosis, rechallenge and treatment selection, and monitoring. This schematic figure underscores the multidimensional utility of liquid biopsy by integrating diverse analytes and specimen types at various decision-making points throughout CRC management.

**Figure 2 ijms-26-07619-f002:**
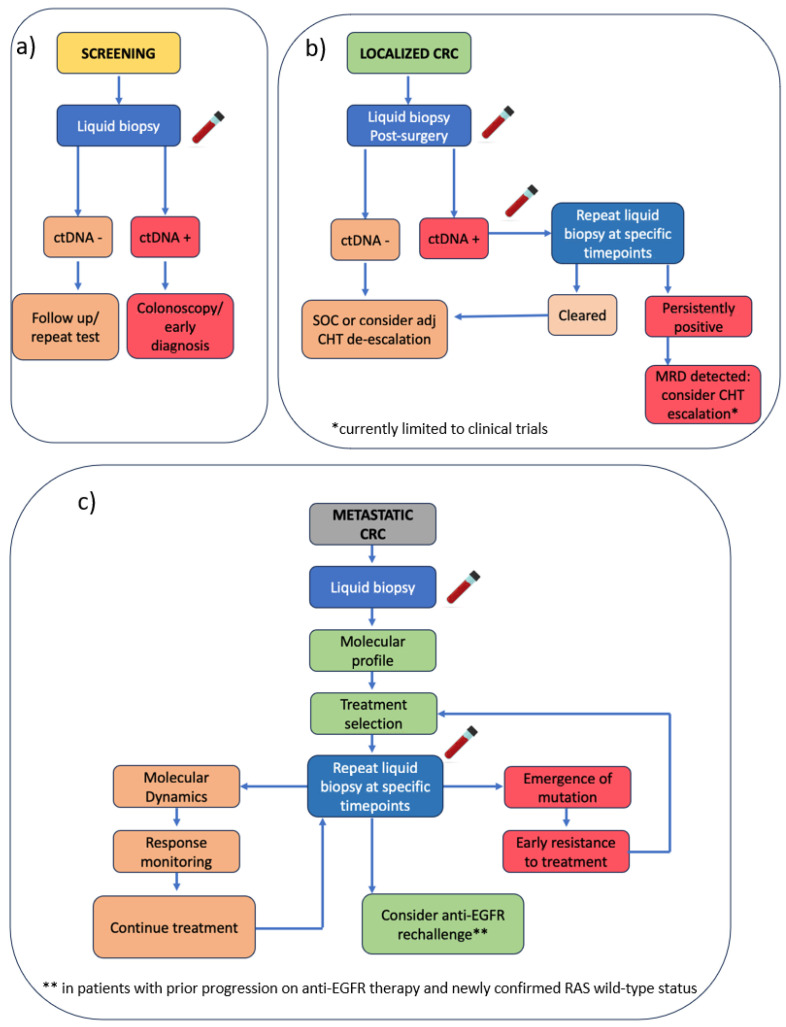
Proposed stage-specific algorithm for integrating liquid biopsy in CRC management. (**a**) Screening: liquid biopsy used for early detection; a positive ctDNA result leads to colonoscopy and potential early diagnosis, while a negative test prompts follow-up or repeat testing. (**b**) Localized CRC: post-surgical ctDNA assessment to guide adjuvant chemotherapy decisions. (**c**) mCRC: liquid biopsy enables molecular profiling to support treatment selection and monitoring response. CRC = colorectal cancer; ctDNA = circulating tumor DNA; MRD = minimal residual disease; CHT = chemotherapy; adj = adjuvant; EGFR = epidermal growth factor receptor.

**Table 1 ijms-26-07619-t001:** Molecular biomarkers in liquid biopsy. CTCs: circulating tumor cells; ctDNA: circulating tumor DNA; ddPCR: digital droplet polymerase chain reaction; qPCR: quantitative polymerase chain reaction; NGS: next-generation sequencing; MRD: minimal residual disease; EpCAM: epithelial cell adhesion molecule; EMT: epithelial–mesenchymal transition; TEPs: tumor-educated platelets; RNA-seq: RNA sequencing; RT-qPCR: reverse transcription quantitative polymerase chain reaction; RNA-FISH: RNA fluorescence in situ hybridization; lncRNAs: long non-coding RNAs; circRNAs: circular RNAs.

Biomarker	Detection Techniques	Advantages	Limitations	References
**CTCs**	Immunofluorescence, EpCAM-based CellSearch^®^, biophysical and immunoaffinity capture	Access to viable tumor cells, multi-omic profiling, metastatic potential assessment	Rare cells, high heterogeneity, EMT reduces detection, lack of standardization	[31,32,33,34,35,36,37,38,39,40,41,42,43,44]
**ctDNA**	ddPCR, qPCR, NGS, Sanger sequencing	Real-time mutation monitoring, MRD tracking, non-invasive, short half-life for dynamic analysis	Low abundance in early disease, fragmented nature, expensive and complex analysis	[45,46,47,48,49,50,51,52,53,54,55,56]
**Exosomes**	Ultracentrifugation, size-exclusion chromatography, microfluidics, immunoaffinity	Stable structure, protected cargo, reflects tumor microenvironment	Isolation complexity, contamination, lack of standard protocols	[57,58,59,60,61,62,63,64,65,66,67,68]
**TEPs**	RNA sequencing (RNA-seq), low-speed centrifugation	Abundant, easy isolation, RNA signatures can indicate tumor features	Mechanisms not fully understood, no FDA-approved assays, standardization needed	[69,70,71,72,73,74]
**miRNAs**	RT-qPCR, RNA-FISH, northern blotting, RNA-seq	Stable in blood, sequence conservation, sensitive detection methods	Sensitive to pre-analytical variables (e.g., hemolysis), standardization issues	[75,76]
**lncRNAs**	RT-qPCR, RNA-seq, hybridization-based methods	Tissue-specific expression, abundant in plasma/serum, epigenetic insights	Function not fully understood, affected by tumor heterogeneity	[77,78,79,80,81,82,83,84]
**circRNAs**	Divergent primer RT-PCR, junction-spanning RNA-seq	High stability, miRNA sponging, transcriptional regulation, specific to tumor types	Early stage research, limited detection standardization	[85,86,87,88,89,90,91,92,93,94,95,96,97]

**Table 2 ijms-26-07619-t002:** Overview of CRC screening methods: conventional and liquid biopsy approaches. ctDNA: circulating tumor DNA; cfDNA: cell-free DNA; miRNA: microRNA; lncRNA: long non-coding RNA; circRNA: circular RNA; FIT: fecal immunochemical test; HSgFOBT: high-sensitivity guaiac-based fecal occult blood test; SEPT9: Septin 9 (gene involved in colorectal cancer); SFRP2: Secreted Frizzled-Related Protein 2; SDC2: Syndecan-2; IGFBP-2: Insulin-like Growth Factor Binding Protein 2; PKM2: Pyruvate Kinase M2; DKK-3: Dickkopf WNT Signaling Pathway Inhibitor 3.

Screening Method	Biomarker Type	Sensitivity/Specificity	Notes	References
Fecal immunochemical test (FIT)	Stool-based	74%/96%	Limited sensitivity, especially for early stages	[99,100]
High-sensitivity guaiac-based fecal occult blood test (HSgFOBT)	Stool-based	70%/93%	Limited sensitivity, especially for early stages	[99,100]
Colonoscopy	Endoscopic	95%/86%	Invasive, poor patient compliance	[99,100]
EpiProColon^®^ 2.0 assay (SEPT9 methylation)	Blood-based (ctDNA)	96%/74%	FDA-approved, limited by small sample size, inconsistent performance	[105,106,107,108]
SpecColon test (SFRP2 and SDC2 methylation)	Blood-based (ctDNA)	Not specified	Limited clinical data	[109]
QuantiDNA™ Test	Blood-based (cfDNA)	Not specified	Non-inferior to colonoscopy but not a primary screening method	[110]
miR-21	Blood-based (miRNA)	77%/83%	Moderate sensitivity, varying results across studies	[111,112,113,114,115,116]
miRNA panel (miR-193a-5p, miR-210, miR-513a-5p, miR-628-3p)	Blood-based (miRNA Panel)	90%/80%	Higher accuracy than individual miRNAs but requires more validation	[117,118,119,120,121]
CCAT1 and HOTAIR (lncRNAs)	Blood-based (lncRNA)	Not specified	Limited data, potential role in early diagnosis	[122,123,124,125,126,127,128]
Exosomal circ_0004771	Blood-based (circRNA)	Not specified	Limited data, potential role in early-stage diagnosis	[129,130,131]
IGFBP-2, PKM2, DKK-3 panel	Blood-based (protein panel)	57%/95%	Panel approach may increase sensitivity but needs further validation	[132,133,134]

**Table 3 ijms-26-07619-t003:** Clinical applications of ctDNA in localized colorectal cancer MRD: minimal residual disease; ctDNA: circulating tumor DNA; HR: hazard ratio; OS: overall survival; DFS: disease-free survival; RFS: relapse-free survival; CRC: colorectal cancer; TTR: time to recurrence; ASCO: American Society of Clinical Oncology.

Application	Study/Trial	Key Findings	Impact	Limitations	Reference(s)
MRD detection after surgery	Reinert et al.	Post-op ctDNA+ → HR 7.2; relapse detected ~8.7 months earlier	Early relapse detection, independent prognostic marker	Small sample for relapse; single-country data	[174]
MRD-guided treatment stratification	CIRCULATE-Japan/GALAXY	DFS worse in ctDNA+; chemo improves outcome (HR 0.4)	Validates ctDNA as predictive marker	Real-world implementation and standardization still evolving	[175,176,177,178]
Landmark analysis of ctDNA dynamics	CIRCULATE-Japan (ASCO 2024 Update)	HR 5.4 for relapse in persistent ctDNA+; 54% achieved sustained ctDNA clearance	Demonstrates significance of ctDNA clearance	Requires further validation in ongoing studies	[178]
Guided adjuvant therapy	DYNAMIC II	3 y RFS non-inferior; less chemo in ctDNA arm	Reduces overtreatment in stage II CRC	Different chemo regimens between groups; underpowered in T4 tumors	[179]
De-/escalation of treatment	DYNAMIC III	2-y RFS: 52% (ctDNA-informed) vs. 61% (SOC); No RFS improvement with CHT escalation; ctDNA burden correlates with relapse risk	Confirms prognostic role of ctDNA; highlights need for new escalation strategies in ctDNA+ stage III CRC.	No survival benefit from treatment intensification; burden-based stratification requires further prospective validation	[180]
De-/escalation of treatment	PEGASUS	FOLFIRI cleared ctDNA in 46%; 7% ctDNA− relapsed	Promising escalation strategy	Small subgroup sizes; recurrence in low-shedding sites	[181]
Feasibility of MRD monitoring	COBRA	Trial halted for high false-positive ctDNA	Highlights preanalytical/technical challenges	Low specificity; assay limitations	[182]
Predictive role of ctDNA for adjuvant FOLFOX + celecoxib	CALGB/ SWOG 80702	ctDNA + predicted worse outcomes; celecoxib improved DFS and OS only in these patients	Highlights ctDNA as a potential tool to guide selective use of celecoxib in stage III colon cancer	Exploratory analysis; benefit observed only in ctDNA+ patients with available biospecimens	[183]
Treatment intensification	Ongoing Trials (CIRCULATE-US, AFFORD, DYNAMIC-III, CLAUDIA)	ctDNA status correlated with recurrence risk	Enhances clinical stratification	Post hoc nature; not primary endpoint of trial	
Detection of occult metastasis	FANTASTIC, AURORA	Early identification of metastases	ctDNA may inform staging decisions	Preliminary data; results pending	[184,185]
Prognostic analysis and adjuvant treatment duration	PRODIGE-GERCOR IDEA-France	2-y TTR: 43.5% in ctDNA+ vs. 88.1% in ctDNA−; ctDNA as independent prognostic marker	Confirms ctDNA as a strong prognostic marker	Limited to stage III CRC; post hoc analysis	[186]
Early intervention in occult metastases	SU2C ACT3 (Pappas et al.)	13.7% ctDNA+ post-chemo; limited actionable biomarkers	Potential for ctDNA-guided early intervention	Low ctDNA positivity rate; protocol modifications	[187]
Surgical treatment planning	IMPROVE-IT2	ctDNA to guide perioperative therapy	May personalize surgical decisions	Protocol phase; clinical endpoints not reported	[188]
Real-world clinical utility	BESPOKE CRC	Real-world outcomes of ctDNA-guided therapy	Implementation in diverse settings	Ongoing; limited peer-reviewed data	[189,190]

**Table 4 ijms-26-07619-t004:** ctDNA in non-metastatic rectal cancer: predictive and prognostic roles. LARC: locally advanced rectal cancer; ctDNA: circulating tumor DNA; OS: overall survival; RFS: relapse-free survival; TNT: total neoadjuvant therapy; pCR: pathological complete response.

Application	Study	Key Findings	Impact	Limitations	Reference(s)
Preoperative treatment stratification	PRODIGE 23	Total neoadjuvant treatment (TNT) improved pathological complete response (pCR) and overall survival (OS)	Confirms TNT as a superior approach in rectal cancer	Data mainly focused on TNT, limited data on ctDNA	[192,193,194]
Predictive role of ctDNA pre-treatment	Appelt et al.	Positive ctDNA associated with worse OS	Confirms ctDNA as a predictive marker for poor outcomes	Small cohort, requires validation in larger studies	[195]
Monitoring ctDNA response to preoperative therapy	Murahashi et al.	ctDNA detected in 57.6% (baseline) and 22.3% (post-TNT); ctDNA clearance associated with higher pCR (*p* = 0.0276)	Demonstrates potential of ctDNA as a marker of response to neoadjuvant therapy	Single-center study; need for multicenter validation	[196]
Prognostic role of ctDNA post-neoadjuvant treatment	Chang et al. (Meta-analysis)	Post-neoadjuvant ctDNA positivity linked to worse RFS (HR 9.16), OS (HR 8.49), and pCR (OR 0.40)	Highlights ctDNA as a key prognostic marker after TNT	Heterogeneity among included studies; varying ctDNA assays	[197]
ctDNA as a prognostic marker	Additional meta-analyses	Consistent evidence supporting ctDNA as a prognostic marker in LARC	Reinforces role of ctDNA as a prognostic biomarker	Varying methodologies and ctDNA detection platforms	[198,199,200]
Postoperative ctDNA monitoring	Tie et al.	Post-op ctDNA+ associated with worse RFS (HR 6.6) and OS (HR 13.0)	Confirms ctDNA as a marker of residual disease and recurrence risk	Retrospective analysis; lack of standardization in ctDNA testing	[201]
Adjuvant therapy stratification	DYNAMIC RECTAL	46% of ctDNA-guided patients received adjuvant chemo vs. 77% in control; 3 y RFS: 76% vs. 82%	Potential to reduce overtreatment through ctDNA-guided approach	Small sample size; ongoing TNT trials not considered	[202]

**Table 5 ijms-26-07619-t005:** Overview of liquid biopsy studies in anti-EGFR therapy for metastatic colorectal cancer. EGFRi: epidermal growth factor receptor inhibitors; ctDNA: circulating tumor DNA; RAS-wt: RAS wild-type; PFS: progression-free survival; OS: overall survival; ORR: objective response rate; BRAF: B-Raf Proto-Oncogene, Serine/Threonine Kinase; MAP2K1: Mitogen-Activated Protein Kinase Kinase 1; VEGFi: Vascular Endothelial Growth Factor Inhibitors; PI3K: Phosphoinositide 3-Kinase; MET: Mesenchymal–Epithelial Transition Factor; BRAF/EGFR: B-Raf Proto-Oncogene, Serine/Threonine Kinase/Epidermal Growth Factor Receptor; GCP: comprehensive genomic profiling.

Study	EGFRi	Type of Study	Focus	Outcome	References
LIBImAb (ASCO 2023)	Cetuximab, Panitumumab	Phase III	ctDNA detection of RAS mutations in RAS-wt patients; 9.5% detected as RAS-mut	Refined patient stratification, potential shift to VEGFi for RAS-mut patients	[258]
FIRE-4	FOLFIRI + Cetuximab	Phase III	13% discordance between tissue and ctDNA RAS status	Highlighting importance of baseline ctDNA testing	[259,260]
Valentino Trial	Panitumumab + 5 fluorouracile	Retrospective Analysis	Identification of molecular alterations during panitumumab maintenance	Identification of poor PFS/OS predictors; ctDNA monitoring recommended	[261]
MODUL (NCT02291289)	Various EGFR inhibitors	Phase II/III	Adaptive therapy based on ctDNA dynamics	Pending results	-
COPERNIC (NCT05487248)	Chemotherapy + EGFR inhibitors	Phase II/III	Evaluating ctDNA-guided therapy in third-line setting	Pending results	-
CRICKET (Phase II)	Cetuximab + Irinotecan	Phase II	Rechallenge with cetuximab + irinotecan; 48% RAS mutations detected	Limited efficacy in RAS-mut patients; RAS-wt responded better	[262]
REMARRY	Panitumumab + Irinotecan	Retrospective Analysis	Rechallenge with panitumumab + irinotecan based on ctDNA RAS-wt reversion	Longer EGFRi-free interval improved outcomes; ctDNA crucial for selection	[263,264]
VELO Trial	Panitumumab + Trifluridine/Tipiracil	Retrospective Analysis	Rechallenge in RAS/BRAF-wt ctDNA patients	Prolonged benefit in ctDNA-confirmed RAS/BRAF-wt cases	[265]
CAVE Study	Cetuximab + Avelumab	Phase II	Combination of cetuximab and avelumab; non-ctDNA guided	OS benefit observed; non-ctDNA guided	[266]
CHRONOS	Panitumumab	Phase II	ctDNA-based ultra-selection for rechallenge with panitumumab	Confirmed value of ctDNA ultra-selection in rechallenge	[267]
REMARRY-PURSUIT	Panitumumab + Irinotecan	Phase II	Assessment of ctDNA RAS dynamics in rechallenge setting	Modest ORR (14%); exclusion of BRAF/EGFR mutations may have impacted outcomes	[263,264]
Mariani et al.	Cetuximab	Retrospective Analysis	Rechallenge strategy based on EGFRi-free interval and ctDNA RAS status	Improved outcomes with >14-month EGFRi-free interval	[268]
CAVE-2 GOIM (NCT05291156)	Cetuximab + avelumab vs. cetuximab	Phase II	Baseline ctDNA CGP (FoundationOne Liquid CDx) in RAS/BRAFV600E-WT, MSS mCRC to guide rechallenge	41.7% had resistance mutations; 52.7% had actionable alterations (ESCAT); ultra-selection based on ctDNA may guide treatment	[269]
Alshammari K et al.	Trametinib + Panitumumab	Phase II	Combination therapy in RAS, BRAF, MAP2K1 mutations; poor tolerability	Trial terminated due to poor tolerability	[270]
PERSPECTIVE	Tepotinib + Cetuximab	Phase II	Targeting MET amplification; limited efficacy	Accrual limitations; promising initial findings	[271]
PARERE (NCT04787341)	EGFR inhibitors	Phase II	Sequencing of pani->rego vs. rego->pani in ctDNA-selected RAS/BRAF wt mCRC	Panitumumab before regorafenib showed superior ORR; ctDNA-guided sequencing supports rechallenge strategy	[272]
C-PRECISE-01 (NCT04495621)	Cetuximab + PI3K inhibitor MEN1611	Phase Ib/II	Targeting PIK3CA mutations	Early phase; targeting actionable mutations via ctDNA	[273]
OrigAMI-1 (NCT05379595)	MET inhibitor Amivantamab	Phase II	Rechallenge in L-sided RAS/BRAF/ EGFR wt mCRC post-EGFRi; impact of EGFRi-free interval	Longer EGFRi-free interval (>8.8 mo) improved ORR (32% vs. 7%), PFS (7.0 vs. 2.8 mo), OS trend (16.1 vs. 10.4 mo)	[274]

**Table 6 ijms-26-07619-t006:** Overview of liquid biopsy on BRAF-targeted therapies in metastatic colorectal cancer. BRAF: serine/threonine-protein kinase B-Raf; BRAF-mut: BRAF-mutated; ctDNA: circulating tumor DNA; CNV: copy number variation; MEK: mitogen-activated protein kinase kinase; MET: mesenchymal epithelial transition factor; MSS: microsatellite stable; OS: overall survival; PFS: progression-free survival; RNF43: ring finger protein 43; VAF: variant allele frequency.

Study	Drugs	Type of Study	Focus	Outcome	References
BEACON	Encorafenib + cetuximab ± binimetinib	Phase III	ctDNA analysis at baseline and progression to identify resistance mechanisms in BRAF-mut patients.	Identified KRAS, NRAS, MAP2K1, and MET amplification as key resistance mechanisms. 60% of patients exhibited resistance alterations.	[305]
Ji et al.	Encorafenib + cetuximab, vemurafenib + cetuximab + irinotecan	Case series	Rechallenge after progression on BRAF inhibitors; monitoring ctDNA for clonal evolution.	One patient responded for 8 cycles, another for 25 cycles; ctDNA VAF predicted treatment response.	[306]
Oddo et al.	Vemurafenib + crizotinib	Case study	Rechallenge using MET-targeted therapy; ctDNA monitoring for BRAF and MET alterations.	Reduction in BRAF VAF and MET CNV anticipated progression 8 weeks prior to imaging.	[307]
Ye et al.	Encorafenib + cetuximab ± binimetinib	Retrospective analysis	Assessment of baseline VAF as a prognostic marker and response indicator.	Patients with lower baseline VAF (<5%) had longer OS and PFS; VAF decline correlated with response.	[308]
RNF43 analysis	BRAF inhibitors	Retrospective analysis	Investigating the role of RNF43 mutations as predictive biomarkers for BRAF-targeted therapy.	RNF43 mutations associated with improved PFS, OS, and response rates in MSS tumors.	[309,310]
VAF analysis	BRAF inhibitors + MEK inhibitors	Retrospective analysis	Correlation of baseline VAF with tumor burden and response to BRAF-targeted therapies.	High baseline VAF (≥2%) correlated with poorer outcomes but higher response to MEK inhibition.	[304,309]
Rechallenge studies	Encorafenib, vemurafenib, cetuximab	Case series	Exploring rechallenge strategies in BRAF-mutant patients post progression.	Patients with persistent BRAF-V600E post progression benefited from rechallenge.	[306,307]

## Data Availability

All data analyzed during this study are included in this published article.

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
