# Peer review of "Navigating the Landscape of Liquid Biopsy in Colorectal Cancer: Current Insights and Future Directions"

_ijms, 2025, doi:10.3390/ijms26157619_

Round 1
Reviewer 1 Report
Comments and Suggestions for Authors
Ziranu et al. presented a well-organized and comprehensive review on the progress of liquid biopsy in colorectal cancer. One of the most impressive aspects of their review is the detailed summary and comparison of various techniques and biomarkers, which are presented in well-organized tables. These tables clearly illustrate the progress and limitations of the corresponding methods. Overall, this high-quality review effectively depicts the current status of liquid biopsy in colorectal cancer. It can be accepted for publishing.
Author Response
Ziranu et al. presented a well-organized and comprehensive review on the progress of liquid biopsy in colorectal cancer. One of the most impressive aspects of their review is the detailed summary and comparison of various techniques and biomarkers, which are presented in well-organized tables. These tables clearly illustrate the progress and limitations of the corresponding methods. Overall, this high-quality review effectively depicts the current status of liquid biopsy in colorectal cancer. It can be accepted for publishing.
R: We sincerely thank the Reviewer for the positive feedback and appreciation of our work.
Reviewer 2 Report
Comments and Suggestions for Authors
This narrative review by Ziranu et al., offers an extensive and timely overview of liquid biopsy applications in colorectal cancer (CRC), spanning early detection, minimal residual disease (MRD) assessment, and treatment monitoring in metastatic settings. The manuscript stands out for its comprehensive literature coverage, clear structure, and detailed discussion of biomarkers such as ctDNA, CTCs, miRNAs, lncRNAs, and circRNAs. The authors succeed in synthesizing recent advances and integrating evolving technologies with current clinical needs, making the review relevant to both researchers and clinicians.
However, some areas require clarification, additional discussion, or structural improvements to enhance the manuscript’s impact and clinical relevance.
Major comments
While the review is impressively thorough, certain sections (on anti-EGFR rechallenge and trials,for example) occupy disproportionate space relative to others, risking imbalance. Consider condensing highly technical parts or moving detailed tables to supplementary material.
The ms largely presents a descriptive summary. A more critical discussion of the quality and limitations of cited studies like retrospective designs, small sample sizes, or lack of standardization—would strengthen the review’s authority and usefulness for clinicians.
Although the title suggests a forward-looking perspective, the “future directions” section is limited. Expand on concrete proposals for research priorities, clinical trial design improvements, or technological innovations (integration of methylation and fragmentomics).
While the authors list numerous biomarkers, a comparative table or figure summarizing their diagnostic accuracy, strengths, and limitations (sensitivity, specificity, current clinical use) would greatly benefit readers.
A table summarizing key biomarkers (ctDNA, CTCs, miRNAs, lncRNAs, circRNAs) with detection techniques, sensitivity/specificity, clinical readiness, and limitations.
The Liquid biopsy implementation in clinical practice entails challenges beyond technical validation, including cost-effectiveness, accessibility, and ethical issues regarding incidental findings. these aspects briefly would contextualize the translational potential.
A concluding figure illustrating an algorithm for integrating liquid biopsy into CRC management, including screening, MRD detection,and metastatic disease monitoring would help.
Minor suggestions
Standardize terminology (e.g., “liquid biopsy platform” vs. “assay”; consistent abbreviations for ctDNA, CTCs, MRD) for clarity.
Some figures (like Fig. 1) could benefit from more clearer description
Overall writing is clear, but several long sentences could be split for easier reading. Minor grammatical errors should be corrected ( “colonoscopy with 95% sensitivity and 86% specificity”, please clarify whether sensitivity/specificity refer to colonoscopy or FIT.
certain trial descriptions include excessive procedural detail (specific assay brands or patient inclusion criteria) that may distract from the main narrative; consider summarizing and referencing original studies for those seeking details..
A Short discussion on regulatory aspects (FDA approvals, current guidelines??) to support clinicians considering implementation.
Author Response
This narrative review by Ziranu et al., offers an extensive and timely overview of liquid biopsy applications in colorectal cancer (CRC), spanning early detection, minimal residual disease (MRD) assessment, and treatment monitoring in metastatic settings. The manuscript stands out for its comprehensive literature coverage, clear structure, and detailed discussion of biomarkers such as ctDNA, CTCs, miRNAs, lncRNAs, and circRNAs. The authors succeed in synthesizing recent advances and integrating evolving technologies with current clinical needs, making the review relevant to both researchers and clinicians.
However, some areas require clarification, additional discussion, or structural improvements to enhance the manuscript’s impact and clinical relevance.
R: We thank the Reviewer for the constructive and thoughtful comments, as well as for the positive assessment of our manuscript. In response to the Reviewer’s suggestions, we have revised the manuscript to clarify key points, expand critical discussion, and improve the overall structure to enhance clarity and impact. Specific responses to each comment are detailed below.
Major comments
While the review is impressively thorough, certain sections (on anti-EGFR rechallenge and trials,for example) occupy disproportionate space relative to others, risking imbalance. Consider condensing highly technical parts or moving detailed tables to supplementary material.
R: We thank the reviewer for the valuable feedback and for recognizing the comprehensive nature of our review. We acknowledge that the section on anti-EGFR rechallenge and related trials was particularly detailed. In response, we have carefully revised and streamlined this section to enhance clarity and conciseness, reducing excessive technical descriptions while preserving the essential clinical and translational content. Rather than moving the tables to supplementary material, we opted to retain them in the main text due to their relevance and utility for readers. However, we have ensured they are referenced more succinctly and clearly within the text. We believe this approach maintains balance across sections while preserving the practical value of the information provided.
The ms largely presents a descriptive summary. A more critical discussion of the quality and limitations of cited studies like retrospective designs, small sample sizes, or lack of standardization—would strengthen the review’s authority and usefulness for clinicians.
R: We thank the reviewer for this valuable comment. In response, we revised the Conclusions section to include a critical discussion of the quality and limitations of the cited evidence, explicitly distinguishing between early-stage and advanced colorectal cancer. For early-stage disease, we highlight that most studies assessing ctDNA are exploratory or single-arm, with limited follow-up and a lack of prospective validation in randomized settings. In contrast, in metastatic disease, although a greater body of evidence exists, many rechallenge studies remain constrained by small cohorts, absence of control arms, and heterogeneous inclusion criteria. We also address the lack of harmonized ctDNA assay platforms and thresholds across studies, which limits cross-trial comparability. We believe these additions improve the critical depth of the review and strengthen its clinical relevance.
Although the title suggests a forward-looking perspective, the “future directions” section is limited. Expand on concrete proposals for research priorities, clinical trial design improvements, or technological innovations (integration of methylation and fragmentomics).
R: We appreciate the reviewer’s suggestion to enhance the forward-looking component of the manuscript. In response, we expanded the Conclusions section to include more concrete proposals regarding future research priorities and clinical applications. Specifically, we now highlight the need for prospective, biomarker-driven randomized trials incorporating serial liquid biopsy assessments to refine treatment algorithms. Additionally, we emphasize the importance of integrating advanced molecular features such as methylation signatures, fragmentomics, and multi-omic profiling to improve sensitivity and specificity, particularly in low-shedding tumors and early-stage disease. We also propose the development of standardized trial frameworks to validate ctDNA-guided therapeutic interventions, including dynamic monitoring and early molecular endpoints. These additions aim to align the conclusions more closely with the translational and innovative focus implied by the title.
While the authors list numerous biomarkers, a comparative table or figure summarizing their diagnostic accuracy, strengths, and limitations (sensitivity, specificity, current clinical use) would greatly benefit readers.
A table summarizing key biomarkers (ctDNA, CTCs, miRNAs, lncRNAs, circRNAs) with detection techniques, sensitivity/specificity, clinical readiness, and limitations.
R: We thank the reviewer for this valuable suggestion. As requested, we confirm that a comparative table (Table 1) is already included in the manuscript. This table provides a structured overview of the main circulating biomarkers in colorectal cancer, including their detection methods, sensitivity and specificity, clinical applicability, and limitations. To improve clarity and accessibility, we have now added an explicit reference to Table 1 in the Section 1.2 (“1.2. Molecular Biomarkers in Liquid Biopsy: Features, Advantages, Limitations, and Detection Techniques”). We hope this adequately addresses the Reviewer’s request. However, we remain available to modify or expand the table further should additional detail be considered necessary.
The Liquid biopsy implementation in clinical practice entails challenges beyond technical validation, including cost-effectiveness, accessibility, and ethical issues regarding incidental findings. these aspects briefly would contextualize the translational potential.
R: We thank the Reviewer for this insightful observation. In response, we added a dedicated paragraph in the Conclusions section to address these critical aspects. The new text briefly discusses cost-effectiveness, accessibility, and ethical implications related to incidental findings, thereby providing a broader perspective on the real-world implementation of liquid biopsy and its translational potential in clinical practice.
A concluding figure illustrating an algorithm for integrating liquid biopsy into CRC management, including screening, MRD detection, and metastatic disease monitoring would help.
R: Thank you for your valuable suggestion. In response, we added a new figure (Figure 2) illustrating a comprehensive algorithm for the integration of liquid biopsy in colorectal cancer (CRC) management. The diagram summarizes the clinical applications across screening, localized disease (MRD detection), and metastatic CRC (monitoring and treatment adjustment), reflecting the concepts and data discussed throughout the manuscript. We believe this addition enhances the clarity and visual synthesis of our review.
Minor suggestions
Standardize terminology (e.g., “liquid biopsy platform” vs. “assay”; consistent abbreviations for ctDNA, CTCs, MRD) for clarity.
R: We thank the Reviewer for the helpful suggestion. We revised the manuscript to ensure consistent terminology and abbreviation usage throughout the text, in line with accepted scientific conventions.
Some figures (like Fig. 1) could benefit from more clearer description
R: We thank the Reviewer for this helpful suggestion. We have revised the legend of Figure 1 to provide a more concise and structured description of its elements. This revision aims to enhance clarity and support readers in interpreting the multidimensional utility of liquid biopsy.
Overall writing is clear, but several long sentences could be split for easier reading. Minor grammatical errors should be corrected (“colonoscopy with 95% sensitivity and 86% specificity”, please clarify whether sensitivity/specificity refer to colonoscopy or FIT.
R: We thank the Reviewer for this valuable feedback. We carefully revised the manuscript to improve readability by splitting overly long sentences and correcting minor grammatical issues. In particular, we clarified the statement regarding screening performance: the sensitivity and specificity values refer to colonoscopy, not to the fecal immunochemical test (FIT). The revised sentence now accurately reflects this distinction.
certain trial descriptions include excessive procedural detail (specific assay brands or patient inclusion criteria) that may distract from the main narrative; consider summarizing and referencing original studies for those seeking details.
R: We thank the Reviewer for this insightful suggestion. We reviewed the manuscript to identify trial descriptions with overly detailed procedural information. Where appropriate, we streamlined these sections to maintain focus on the main narrative and clinical implications, while referencing the original studies to allow interested readers to access specific methodological details.
A Short discussion on regulatory aspects (FDA approvals, current guidelines??) to support clinicians considering implementation.
R: We thank the Reviewer for this important observation. In response, we added a dedicated paragraph in the Conclusions section to address regulatory aspects related to the implementation of liquid biopsy in clinical practice. This addition briefly outlines current FDA approvals (e.g., Breakthrough Device Designation for MRD assays such as Signatera™) and summarizes recommendations included in the latest NCCN and ESMO guidelines for metastatic CRC and MRD monitoring. These updates aim to provide clinicians with a clearer understanding of the translational readiness and current regulatory landscape supporting liquid biopsy integration.
Reviewer 3 Report
Comments and Suggestions for Authors
I find this to be an excellent review of liquid biopsy approaches as applied to colon cancer. The authors have broadened the coverage to well beyond CTC and ctDNA and covered work using exosomes, different circulating RNAs, and tumor educated platelets. These approaches are summarized in an informative table with references and giving the advantages/disadvantages and the methods used. This is the first of a number of useful tables in the review. The historical development of liquid biopsy is described, something I have not seen elsewhere. Screening methods are covered in detail as there is great interest in future alternatives to invasive colonoscopy. The full range of biomarkers relevant to colon cancer is covered. The use of liquid biopsy for different disease stages and followup is covered in detail for both metastatic and non-metastatic cases. Ongoing clinical trials for liquid biopsy for anti-EGFR and those where the BRAF mutant is targeted are summarized in detail with tables. A discussion of new and emerging biomarkers concludes the article. It is a very valuable review.
Author Response
I find this to be an excellent review of liquid biopsy approaches as applied to colon cancer. The authors have broadened the coverage to well beyond CTC and ctDNA and covered work using exosomes, different circulating RNAs, and tumor educated platelets. These approaches are summarized in an informative table with references and giving the advantages/disadvantages and the methods used. This is the first of a number of useful tables in the review. The historical development of liquid biopsy is described, something I have not seen elsewhere. Screening methods are covered in detail as there is great interest in future alternatives to invasive colonoscopy. The full range of biomarkers relevant to colon cancer is covered. The use of liquid biopsy for different disease stages and followup is covered in detail for both metastatic and non-metastatic cases. Ongoing clinical trials for liquid biopsy for anti-EGFR and those where the BRAF mutant is targeted are summarized in detail with tables. A discussion of new and emerging biomarkers concludes the article. It is a very valuable review.
R: We thank the Reviewer for the positive and encouraging feedback.
Round 2
Reviewer 2 Report
Comments and Suggestions for Authors
The authors have adequately addressed the concerns and made significant changes to their review manuscript, including the addition of new information. However, these additions have made the manuscript somewhat heavy to read. The authors should try to be more concise wherever possible.
Author Response
The authors have adequately addressed the concerns and made significant changes to their review manuscript, including the addition of new information. However, these additions have made the manuscript somewhat heavy to read. The authors should try to be more concise wherever possible.
R: We thank the reviewer for the positive feedback and thoughtful suggestion. We fully agree that the inclusion of new data and expanded discussion, while necessary to ensure completeness, may have contributed to reduced readability. In response, we have carefully revised the manuscript to improve clarity and conciseness. We hope the revised version meets the reviewer’s expectations.